# Development and Validation of an Autonomous System for Measurement of Sunshine Duration

**DOI:** 10.3390/s20164606

**Published:** 2020-08-16

**Authors:** Álvaro B. da Rocha, Eisenhawer de M. Fernandes, Carlos A. C. dos Santos, Júlio M. T. Diniz, Wanderley F. A. Junior

**Affiliations:** 1Academic Unit of Mechanical Engineering, Federal University of Campina Grande, Campina Grande, Paraíba 58429-900, Brazil; alvarobarbosa2@hotmail.com (Á.B.d.R.); eisenhawer@ee.ufcg.edu.br (E.d.M.F.); wanderley.ferreira@ufcg.edu.br (W.F.A.J.); 2Academic Unit of Atmospheric Sciences, Federal University of Campina Grande, Campina Grande, Paraíba 58429-900, Brazil; 3School of Apprentices-Sailors of Pernambuco, Navy of Brazil, Recife, Pernambuco 53110-901, Brazil; julio.tavares@marinha.mil.br

**Keywords:** autonomous electronic system, sunshine duration, fuzzy logic, low-cost solution

## Abstract

This paper presented an autonomous electronic system for sunshine duration (SD) monitoring based on the contrast method and developed to operate on a horizontal surface. The prototype uses four photoresistors arranged at 90° in a 20 mm diameter circumference separated by a shading structure used to create a shadow pattern on the detection element. Photoresistors are inserted in individual signal conditioning circuits based on the association between Wheatstone bridges and operational amplifiers to provide an analog signal to the microcontroller unit. The determination of SD occurs through the implementation of fuzzy logic with numerical calculation methods to estimate the probability (f) of solar disk obstruction and estimate SD values. The system does not require additional adjustments after installation or use of energy sources for operation due to the use of an internal battery with charge recovery by solar panels. Experimental results of the proposed system were validated with the ones provided by a government meteorology station. Statistical analysis of the results showed a confidence index (c) greater than 90%, with a precision of 94.26%. The proposed system is a feasible low-cost solution to the available commercial systems for the measurement of sunshine duration.

## 1. Introduction

Knowledge of the sunshine duration (SD), also called insolation, indicates the total length of the period in which clouds do not block direct solar radiation at a given location on the Earth’s surface [1,2,3,4]. SD measurement is an important parameter for meteorology, health, agriculture, architecture, and photovoltaic generation systems.

The measurement of SD can be carried out by using direct measuring instruments such as the Campbell–Stokes heliograph, pyrheliometer, and digital sensors, or by indirect measuring instruments such as a pyranometer. These measurement devices require positioning in a plane level with the horizon that is free of possible obstacles that act as sources of interference, providing shading on the device [5].

The Campbell–Stokes heliograph or interim reference sunshine recorder (IRSR) is an apparatus for the direct measurement of SD with greater acceptance and use in meteorological stations [2,6,7]. The heliograph features a glass sphere lens that focuses the direct solar irradiance of the sun under a graduated paper tape, burning it as the solar disk clears [4,8,9].

The interruption of the burnt strip indicates the presence of clouds, and the SD calculation is carried out by an on-site observer who reads the burnt paper tape, thus determining the number of hours. Nevertheless, the Campbell–Stokes heliograph has low accuracy for periods of high cloudiness or periods of intermittent sun exposure, leading to errors in sunshine values [6,10,11].

Pyrheliometers are used to determine the SD by measuring the direct component of solar radiation by thermal detectors attached to a tungsten trioxide (WO3) surface [5,6]. They have a sensitivity angle to capture solar radiation with a displacement of less than 30°, so the device requires adjustments with the variation of the solar disk’s position.

The digital sensors used to determine the SD use, in short, most of the shadow formation method, or method of contrast, to measure global and diffuse components of solar radiation. These sensors employ silicon-based devices to identify variations higher than 120 W m^−2^ between the components of solar radiation [5,11,12]. The devices that use the contrast method, unlike the IRSR and the pyrheliometer, do not need additional adjustment after being positioned on a horizontal base [5].

The use of pyranometers in determining SD requires the use of statistical-mathematical models such as the Angstrom–Prescott model, which relates the ratio between the solar energy incident on the surface and the extraterrestrial energy with the ratio between the observed daily SD and the photoperiod estimated [5,13,14].

The use of statistical-mathematical models is a valuable tool only when accompanied by a radiometric measurement database associated with the knowledge of the values for the regression coefficients [4,6,15]. The determination of SD by mathematical models is considered restricted and not feasible for most locations due to the lack of consistent meteorological data necessary for the elaboration of such models [8,16,17].

The SD measurement, regardless of the method used, constitutes a high cost of acquisition and operation, making it impossible in most of the Brazilian and worldwide meteorological stations. In China, endowed with one of the largest climate monitoring networks in the world, it carries out observations of sunlight in approximately 16% of the stations that integrate the national meteorological observation network [18].

Alternative methods for determining the SD are based on the use of the Campbell–Stokes heliograph. This method is an extensively popular and accepted tool for recording the SD despite errors accumulated throughout the measurement process [13,19,20].

The difficulties and high values of acquisition, operation, and physical structure dedicated to the measurement of an SD as well as other environmental variables have led to the development of alternatives with greater commercial and practical feasibility such as the use of automatic measuring instruments, satellite image mapping model, and artificial neural networks (ANN) to real-time monitoring [21,22].

The purpose of this study is to present a compact, low-cost, autonomous electronic device dedicated to the real-time measurement of the SD requiring a minimal structure. The experimental tests were carried out in the city of Campina Grande, Paraiba, Brazil. In order to validate the developed solution, the data obtained by the proposed device were compared with the official data released by the Brazilian National Meteorological Institute (INMET).

## 2. Materials and Methods

### 2.1. System Design

The proposed system for the measurement of SD is presented in Figure 1. The prototype is composed of four luminosity sensors using a shading pattern that is oriented to the geographic north as a method of measuring SD. The proposed SD measurement system was implemented in a low-power ATmega8535 AVR 8 MHz microcontroller with an internal 10-bit resolution analog-to-digital converter (ADC). Figure 2 shows the hardware structure of the proposed autonomous system.

The luminosity sensors used in this solution were four photoresistor elements based on cadmium sulfide (CdS). Each sensor had a sensitivity area of 2.5 cm^2^ and presented the variation of internal resistance according to the change in incident light levels [23,24]. The use of photoresistor elements has been used to identify the difference between local luminosity instead of directly measuring the levels of irradiance [25]. Furthermore, these sensors were mounted on the top of the prototype.

Each photoresistor element has its conditioning circuit for the output signal. The conditioning circuit is presented in Figure 3 and consists of a Wheatstone bridge, a differential amplifier (U1), and an active low-pass filter (U2). The resistors used in the Wheatstone circuit of R1, R2, and R3 were 100 Ω, 1 Ω and 100 Ω, respectively. The U1 amplified the output voltage from the Wheatstone bridge with a gain of 3.2.

The U2 has a cut-off frequency of 700 kHz and provides an output voltage of 3865 mV ± 50 mV. The output of the signal conditioning system (Vout) is connected to the microprocessor through the analog-digital converter (ADC) with a resolution of 4.88 mV/bit. The system uses an LCD for the user interface to display data and configure parameters, as shown in Figure 4, and the LCD interface is installed in the front of the prototype. It presents the results of the total or partial period of SD. It also exhibits status messages related to success or errors in data storage. This is performed through a 16 × 2 LCD with I2C. Figure 4 illustrates two examples of data displayed.

The system has an internal datalogger that allows the storage of the insolation data, thus avoiding data losses. The sampling frequency was 1 Hz, receiving the information sent from the microcontroller. The data can be stored for a period of up to two years. The datalogger uses ASCII standard code for write data and has a removable flash memory composed of a MicroSD card.

The system uses two lithium polymer (LiPo) batteries with a combined capacity of 2000 mAh, providing up to 20 h of operation. The battery bank is recharged by local irradiance using two solar panels with a power rating of 5 W. This circuit guarantees the system’s autonomy. The system integrates IC TP4056 to control the charging rate of battery banks at a constant current–voltage rate. Figure 5 presents the circuit used to manage the battery bank charging from solar panels.

### 2.2. Contrast Pattern

The contrast pattern used in the prototype consisted of four photoresistor elements arranged at 90°, separated by a screen that allows the rapid transition of the light state from the isolation of the sensors, regardless of the prototype’s geographic position in relation to the path of the prototype solar disk. The pattern of contrast is shown in Figure 6.

In Figure 6, the contrast pattern was used to create shadows on the sensors positioned on the structure from the movement of the sun. Similar contrast patterns have been approached, such as those developed by Breniuc [5] by using a large number of sensors. However, these solutions do not allow a rapid transition of light due to the use of screens that partially isolate the sensors, resulting in the use of a high sampling rate by part of the sensors. The shading model used in the prototype allows the identification of variations in the luminosity levels used to determine the local sunshine levels, using the algorithm described in Section 2.3.

### 2.3. Algorithm Calculation of Sunshine Duration

Figure 7 shows the flowchart of the algorithm for the calculation of SD. The algorithm was implemented in the ATmega8535 microcontroller using C language and floating-point variables of 4 byte lengths.

The algorithm uses a Boolean variable (*E*) that changes its state according to the microcontroller’s identification of four sensor voltages. The variable *E* assumes 1 only if the sensor voltages are positive, starting with the fuzzy logic and the timer (t1). In contrast, if the sensor voltage is negative or zero, variable *E* assumes zero. This procedure allows for the estimation of the SD regardless of the position of the solar disk in the sky and the moment of sunrise occurrence [26,27]. Figure 8 shows the structure of the implemented fuzzy logic algorithm.

The output value (*f*) describes the probability of the obstruction of the solar disk, and it is integrated for a time step (Δt) of 60 s. The use of discrete-time intervals allows for the determination of the fraction of time that the solar disk remains free of obstructions (i.e., there are no clouds in the sky [14]). The total SD is obtained by the sum of all observed intermittent intervals.

The algorithm uses voltage levels (mV) of all luminosity sensors as input variables, returning the output value (*f*). In the fuzzification stage, four membership functions were defined for each input variable of the fuzzy algorithm and chosen to be trapezoidal functions. The set of membership functions is depicted in Figure 9. In the algorithm, the variables are described based on the linguistic terms defined as “Very high”, “High”, “Medium”, and “Low”. The linguistic terms presented behave as linear pertinence functions by parts, acting as fuzzy subsets in the unit interval [0, 1]. The pertinence functions were established experimentally by comparing the voltage level of the sensors with the formation of shadows and the consequent interruption of the burning of the graduated tape on the Campbell–Stokes heliograph, which was subsequently adjusted empirically, according to the system analyzed. The experiments to develop the pertinence functions were carried out from 8–25 August 2017, at the Brazilian Agricultural Research Corporation (Embrapa).

Regarding the input variables, four membership functions were also defined for the output value (*f*). These functions are trapezoidal, based on experimental knowledge, as shown in Figure 10. Based on this, the linguistic terms were defined as ‘Very High’, ‘High’, ‘Medium’, and ‘Low’. The range of *f* is [0, 1]. If variable *f* assumes 1, this indicates that the sky is partially cloudy or cloudy. On the other hand, if variable *f* is zero, this means that the sky is cloud-free.

At the defuzzification stage, the linguistic rules establish a link between output *f*, membership functions, and its level of membership. The numerical value for f is calculated at this stage. There are several methods for this calculation in the technical literature. In this study, we implemented the centroid defuzzification technique [28,29].

In contrast to the solution proposed by Breniuc [5], which involves the use of the sensor voltage as the input variable instead of using the relationship between the maximum and minimum luminosity levels for the detection of the light state transition, the proposed SD calculation algorithm becomes independent of the geographical characteristics of the region where the system is used. Consequently, altitude, latitude, and longitude do not need to be used as parameters for the algorithm.

### 2.4. Experimental Tests and Prototype

The experimental tests were carried out at the Department of Mechanical Engineering of Federal University of Campina Grande (UFCG), Campina Grande, located in the Northeast Region of Brazil (NEB) with an altitude of 512 m and a tropical climate [30]. The tests were performed in the period between October 2017 and July 2018 and based on the methodology described by Matuszko [3] and Rocha [31] to compare the daily insolation data.

Figure 11 presents the prototype for the SD measurement system in the laboratory, with an estimated cost of US$150.00. The prototype weighed 650 g with the following physical dimensions: length 15.4 cm (L), width 12.4 cm (W), and height 8.7 cm (H).

### 2.5. Data and Experimental Area

The official data for the SD were obtained from a meteorological station located at Embrapa, 7.2258168 S,35.906142 W, situated in the city of Campina Grande, Brazil. The station is equipped with an interim reference sunshine recorder (IRSR), model Campbell–Stokes. The data are officially published online by INMET at 8 pm, UTC-3 [32].

### 2.6. Data Analysis

The statistical analyses were based on the methodologies proposed by Cunha et al. [33], Miot [34], and Diniz [35], which determine statistical parameters such as for mean absolute error (E_ma_), maximum absolute error (E_max_), and Pearson’s correlation coefficient (r), which measure the degree of correlation between the experimental and official data, correlation index (r^2^), which measures the degree of fit of the linear statistical model, and confidence index (c) to estimate the reliability of the experimental data [36]. The classification values for r and c are described in Table 1 [33,35].

In addition to the definition of the above-mentioned statistical parameters, the behavior of the relative frequencies of the individual insolation differences in the form of a histogram was also analyzed.

## 3. Results and Discussion

The performance of the proposed SD measurement system was compared with the data provided by the INMET for 83 days. Figure 12 shows the results between the SD measurements by the prototype with those provided by the official data (INMET).

The device provides accurate measurements of the SD values of a solar angle between 23° and 159°. The measurement of sunlight for values of the solar angle outside the specified range, even on days with low cloudiness, was inefficient due to the sensitivity of the photoresist elements to low radiation levels. Figure 13 illustrates the behavior of the daily SD readings for five observations.

The analysis of results indicates small divergences between the curves, showing a good tracking response of the system. An analysis of the discrepancies was performed using the histogram in Figure 14 to analyze the characteristics of the distribution of the absolute frequencies of the divergences.

The characteristics of the frequency distribution of the divergences were modeled according to a normal distribution of the mean (µ) −0.11446 h and standard deviation (δ) 0.45042 h. The behavior of the normal distribution indicated that 79.51% of the divergences were located in the range between −0.5 h and +0.5 h. Based on the behavior of the normal distribution obtained by the analysis of discrepancies, it is possible to estimate that for any measurement carried out by the prototype, the probability of this reading showing a divergence greater than 0.5 h, in relation to the official data, is 28.23%.

Furthermore, the observed differences were characterized by E_max_ of 2.3 h and E_ABS_ of 0.378 h per day. The behavior of the observed differences, based on the absolute maximum and average errors, were strongly influenced by the high cloudiness index, which induces the Campbell–Stokes heliograph to measure only solar radiation levels above 280 W m^−2^ [12,37,38]. The degree of correlation between the data is shown in Figure 15.

The experimental results presented a degree of fit to the linear statistical model (r^2^) of 0.8885. The correlation coefficient (r) and the confidence index for the system were 0.942 and 0.956, respectively. Thus, according to the coefficient classification presented in Table 1, the results provided by the prototype can be classified as “Almost Perfect” and “Excellent”.

## 4. Conclusions

This paper proposed a new electronic system for sunshine duration (SD) measurement based on a shading pattern to detect the rapid transition of the state of light. The proposed system consists of four photoresistor elements, a signal conditioning system, an ATmega8235 microcontroller, an internal datalogger, and a user-interface to display the total or partial SD. The proposed system is low-cost and has an autonomy of up to 20 h by recharging the batteries through photovoltaic panels.

For the validation of the proposed system, the experimental results were compared with those provided by the INMET for the city of Campina Grande, Brazil. The experimental results of the proposed solution had an accuracy greater than 90%. This result is strongly associated with the proposed shading pattern of the sensor with the fuzzy logic algorithm, allowing for the prototype to detect intermittent variations in the levels of solar radiation.

The shading method used in this work was similar to the one proposed by Breniuc [5]. However, the proposed prototype requires fewer photoresist elements in the sensor composition than those projected by Breniuc [5] and Barros [25]. In addition, the prototype’s response has no dependency on the geographical parameters of the region. Moreover, the proposed system is a feasible alternative candidate to measure and estimate SD adequately with an estimated cost of US$150.00. On the other hand, commercial devices with similar technology such as SPN1 can cost more than US$6000.

## 5. Patents

This device has been registered with the National Institute of Industrial Property under the numbers BR1020180733214 and BR1020180733230.

## Figures and Tables

**Figure 1 sensors-20-04606-f001:**
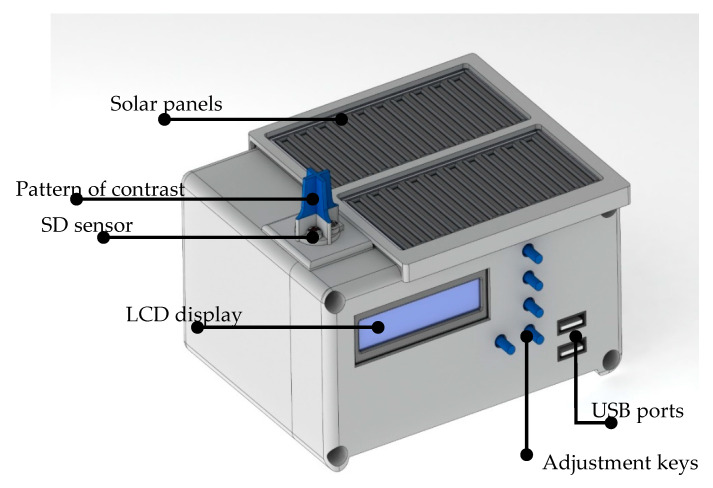
Proposed autonomous system for the measurement of sunshine duration (SD).

**Figure 2 sensors-20-04606-f002:**
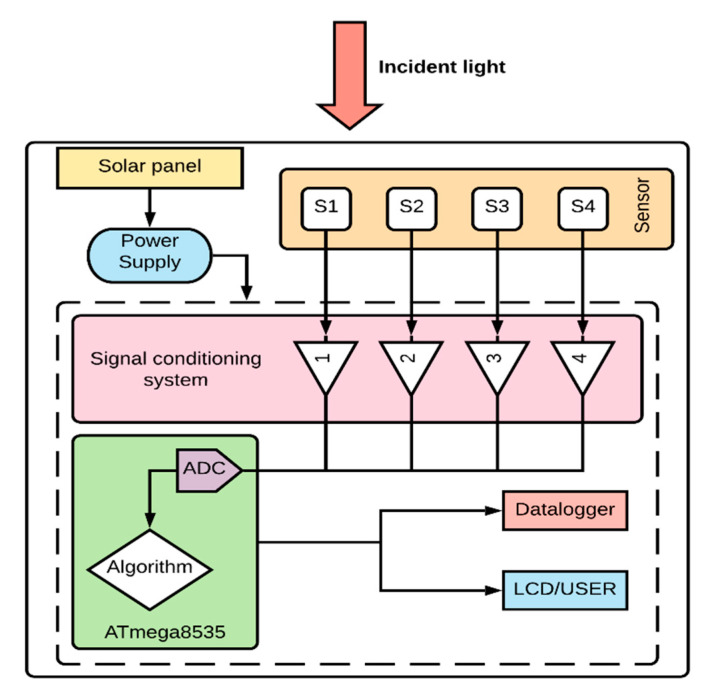
Diagram of the proposed sunshine duration (SD) monitoring system.

**Figure 3 sensors-20-04606-f003:**
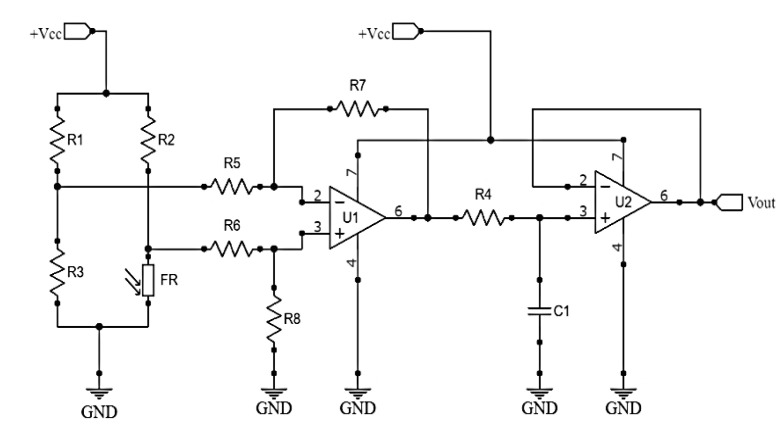
Conditioning circuit for the photoresistor sensors.

**Figure 4 sensors-20-04606-f004:**
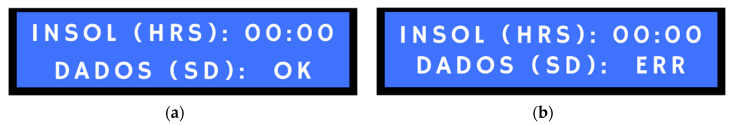
The user interface: (**a**) situations of compliance; (**b**) situations of error in data storage.

**Figure 5 sensors-20-04606-f005:**
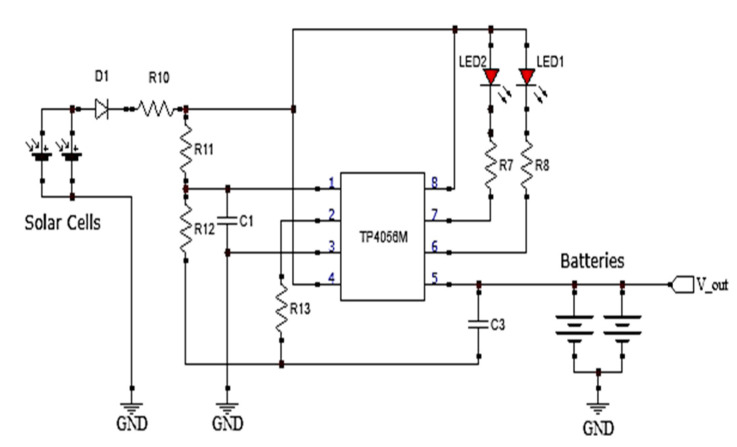
Circuit for battery charging.

**Figure 6 sensors-20-04606-f006:**
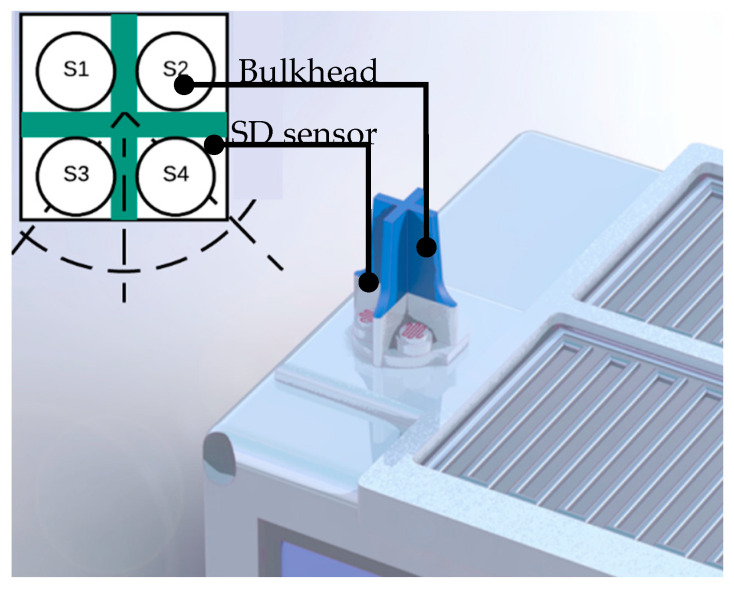
Arrangement of sensors based on the shading pattern.

**Figure 7 sensors-20-04606-f007:**
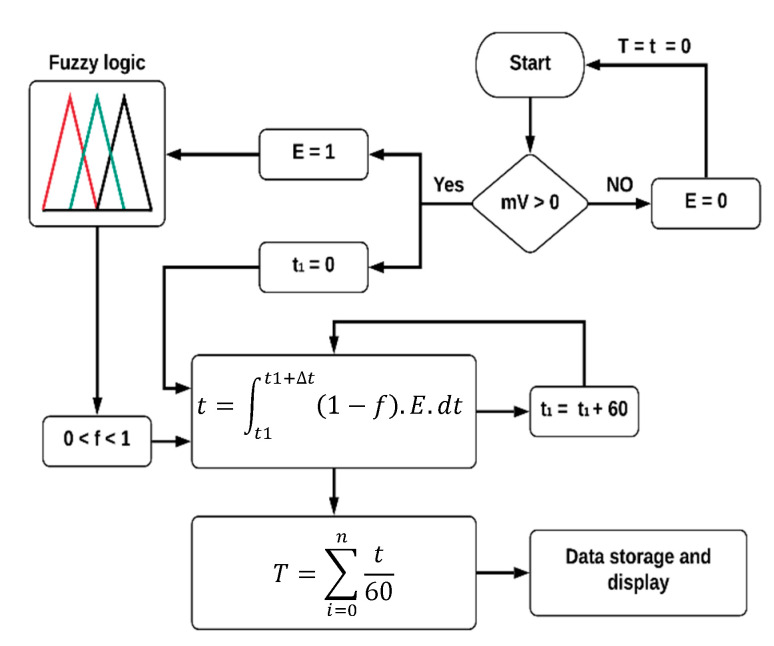
Algorithm for the sunshine duration (SD) calculation.

**Figure 8 sensors-20-04606-f008:**
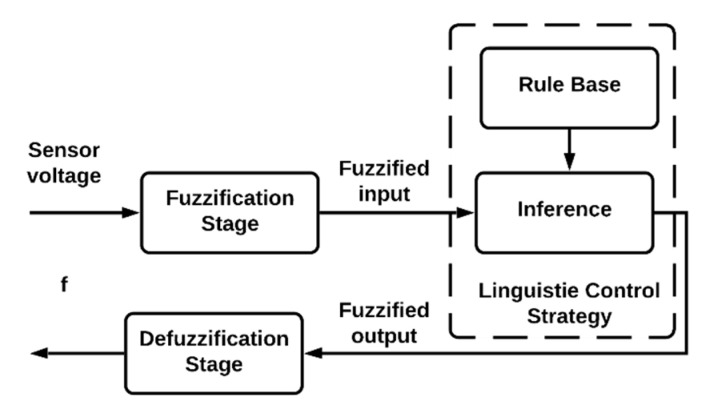
Structure of fuzzy logic algorithm for SD calculation.

**Figure 9 sensors-20-04606-f009:**
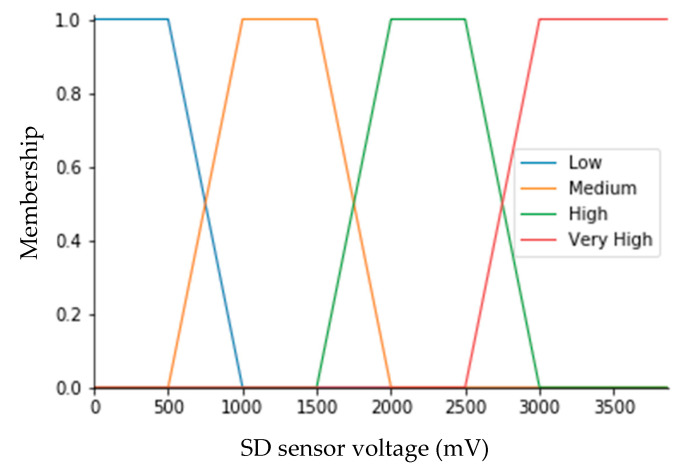
Membership functions for sunshine duration (SD) sensor voltage.

**Figure 10 sensors-20-04606-f010:**
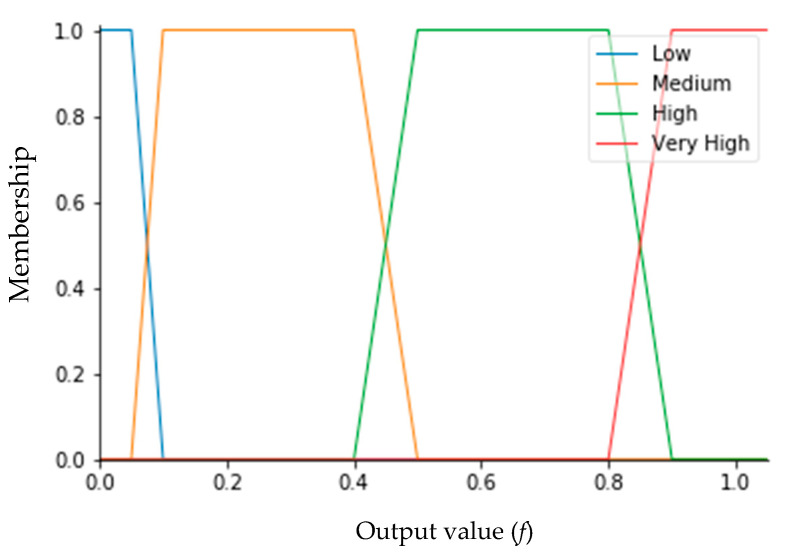
Membership functions for sunshine duration (SD) sensor voltage.

**Figure 11 sensors-20-04606-f011:**
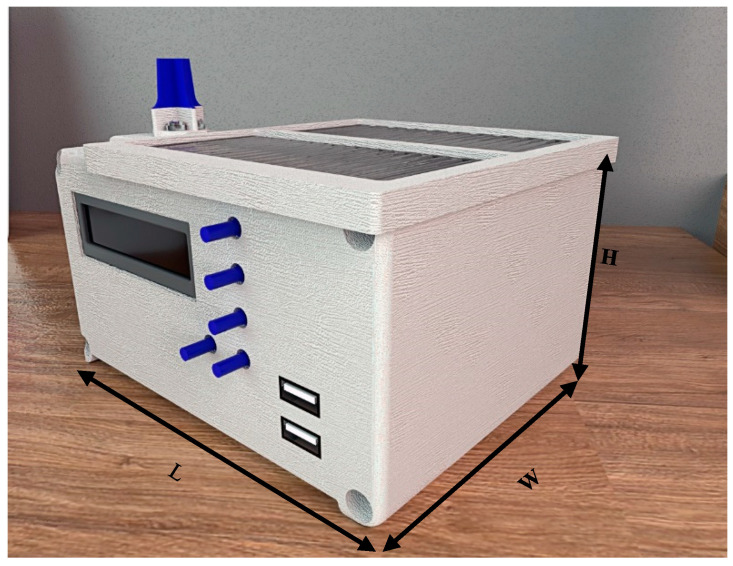
Proposed prototype for measurement of sunshine duration (SD).

**Figure 12 sensors-20-04606-f012:**
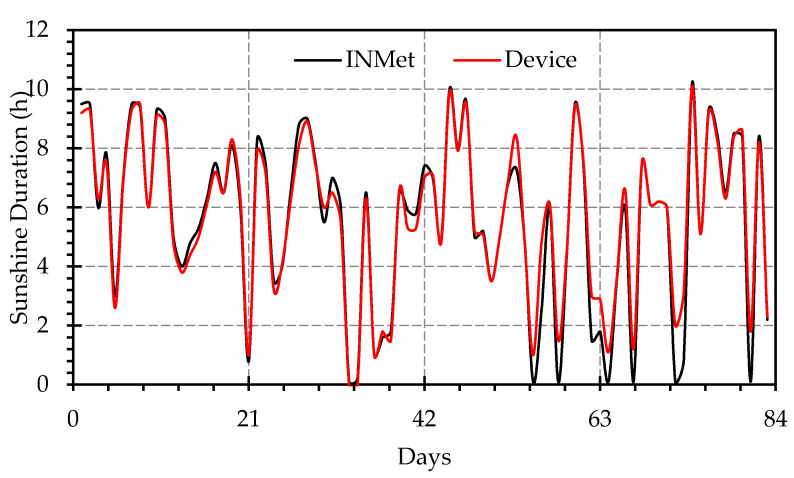
Comparison between prototype response (red line) and official data (black line).

**Figure 13 sensors-20-04606-f013:**
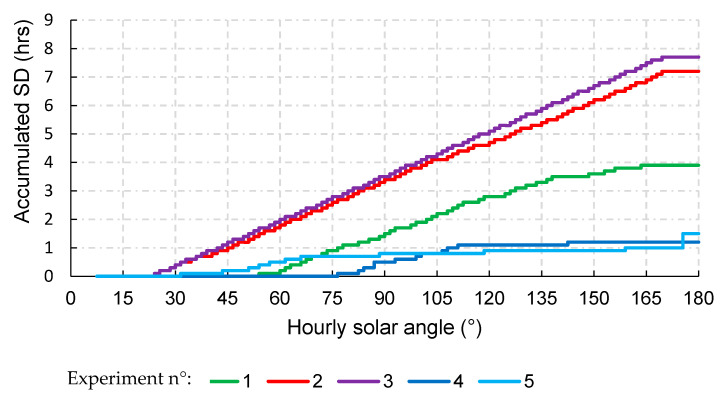
Behavior of daily sunshine readings as a function of the solar hour angle for the device.

**Figure 14 sensors-20-04606-f014:**
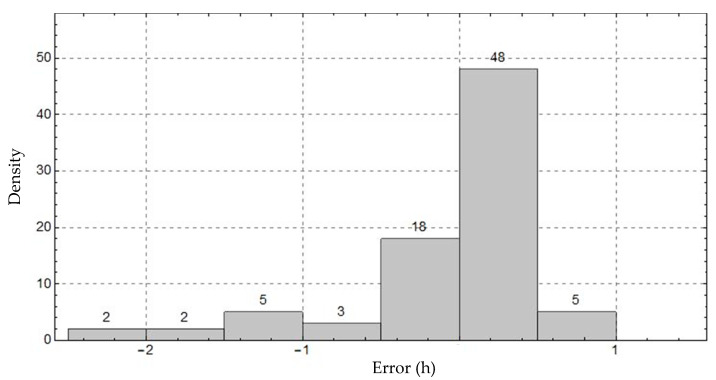
Relative frequency of divergences of the sunshine duration (SD) results between the proposed system and official data for the city of Campina Grande, Brazil.

**Figure 15 sensors-20-04606-f015:**
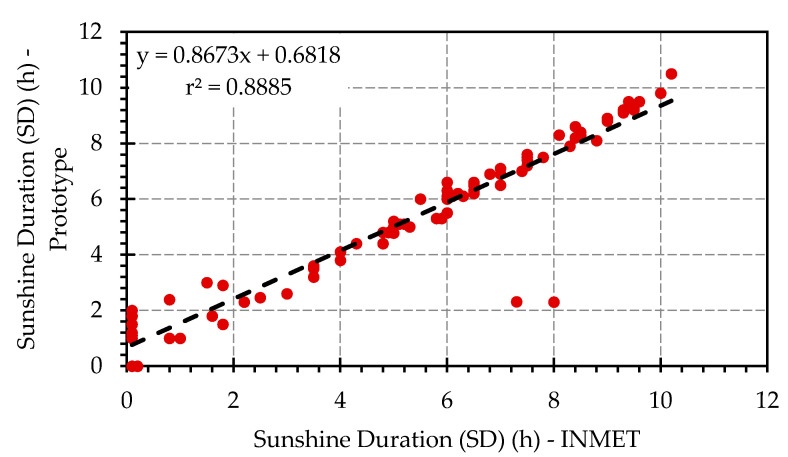
Relationship between sunshine duration (SD) measured with the proposed system and official data by the Brazilian National Meteorological Institute (INMET) for the city of Campina Grande, Brazil.

**Table 1 sensors-20-04606-t001:** Performance classification for Pearson’s correlation coefficient (r) and confidence index (c) [33,35].

Correlation Coefficient (r)	Classification	Confidence Index (c)	Classification
>0.99	Perfect	>0.85	Excellent
0.91–0.99	Almost perfect	0.76–0.85	Very good
0.71–0.90	Very high	0.66–0.75	Good
0.51–0.70	High	0.61–0.65	Satisfactory
0.31–0.50	Moderate	0.51–0.60	poor
0.11–0.30	Low	0.41–0.50	Bad
<0.10	Very low	<0.40	Very bad

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
