# Peer review of "Development and Validation of an Autonomous System for Measurement of Sunshine Duration"

_sensors, 2020, doi:10.3390/s20164606_

Round 1
Reviewer 1 Report
This study seeks to present a compact, low-cost, autonomous electronic device
80 dedicated to real-time measurement of an SD requiring a minimal structure. The paper is very well written, detailing the process. Even already patented. I congratulate the authors for their research and that they have a lot to contribute to the most diverse areas that make use of sunlight and its diverse applications.
Author Response
Dear Editor,
This document summarizes the modifications requested by the reviewers. The line indications and changes signaled in this file are related to the reviewed version.
- The authors present a contrast method in order to determine the sunshine duration, based on a quadrant of 4 photoresistor elements separated by orthogonal screens. I realise that this instrument has been deployed in Brazil, but it would be good if the authors could discuss the more global applicability of this instrument and method. As far as I can tell from the instrument setup and design, it might be problematic to deploy this instrument in high latitude regions, particularly in the winter months as the screens would likely completely shade the detectors when the solar angle is low. The authors could address this issue with data from early morning / late afternoon.
- Line 240
Paragraph added: The device provides accurate measurements of SD values of the solar angle between 23° and 159°. The measurement of sunlight for values of the solar angle outside the specified range, even on days with low cloudiness, is inefficient due to the sensitivity of the photoresist elements to low radiation levels. Figure 13 illustrates the behavior of daily readings of SD for 5 observations.
- Line 244
Image added:
Figure 13. Behavior of daily sunshine readings as a function of the solar hour angle for the device.
- Line 251:
Change in image numbering in the text
Replace of “Figure 13”
Addition of “Figure 14”
- Line 252:
Change in image numbering inte the caption
Replace of “Figure 13”
Addition of “Figure 14”
- Line 268:
Change in image numbering in the text
Replace of “Figure 14”
Addition of “Figure 15”
- Line 269:
Change in image numbering in the caption
Replace of “Figure 14”
Addition of “Figure 15”
- There are other commercially available instruments which use a differential shading pattern, such as the SPN1 (https://www.delta-t.co.uk/product/spn1/) with associated publications which the authors may want to refer to (e.g. Wood et al., 2017 - http://plymsea.ac.uk/id/eprint/7456/6/amt-10-1723-2017.pdf). These would provide a good contrast with the authors instrument on grounds of cost, but that they provide a similar type of technique and robustness.
Replace of “In contrast, commercial devices may cost more than US$ 2,000.00.“
Addition of “In the other hand, commercial devices of similar technique, such as SPN1, can cost more than US $ 6,000.00.”
- Line 174: More details are required about the linguistic guidelines and how these were determined, rather than by saying that they were determined “experimentally”. This could mean anything.
Replace of “In the algorithm, the variables are described based on the linguistic terms defined as ‘Very High’, ‘High’, ‘Medium’ and ‘Low’. These guidelines were determined experimentally.“
Addition of “In the algorithm, the variables are described based on the linguistic terms defined as "Very high", "High", "Medium" and "Low". The linguistic terms presented behave as linear pertinence functions by parts, acting as fuzzy subsets in the unit interval [0.1]. The pertinence functions were established experimentally by comparing the voltage level of the sensors with the formation of shadows and the consequent interruption of the burning of the graduated tape on the Campbell-Stokes heliograph, being subsequently adjusted empirically, according to the system. analyzed. The experiments to develop the pertinence functions were carried out between August 8 and 25, 2017 at the Brazilian Agricultural Research Corporation (Embrapa).”
- Line 25: it appears that Willmott’s confidence index is only mentioned here. It could be shortened to confidence index.
Replacement of “show a Willmott’s confidence index”
Addition of “shows a confidence index”
- I was entertained by the descriptions given in Table 1 as “sufferable” “bad” and “terrible”. However, I am not sure that such qualitative terms really belong in a scientific paper(!)
Change in the classification of the confidence index (c) in Table 1.
Replace of:
|
Correlation coefficient (r) |
Classification |
Confidence index (c) |
Classification |
|
>0.99 |
Perfect |
>0.85 |
Great |
|
0.91 – 0.99 |
Almost perfect |
0.76 – 0.85 |
Very good |
|
0.71 – 0.90 |
Very high |
0.66 – 0.75 |
Good |
|
0.51 – 0.70 |
High |
0.61 – 0.65 |
Median |
|
0.31 – 0.50 |
Moderate |
0.51 – 0.60 |
Sufferable |
|
0.11 – 0.30 |
Low |
0.41 – 0.50 |
Bad |
|
< 0.10 |
Very low |
< 0.40 |
Terrible |
Addition of:
|
Correlation coefficient (r) |
Classification |
Confidence index (c) |
Classification |
|
>0.99 |
Perfect |
>0.85 |
Excellent |
|
0.91 – 0.99 |
Almost perfect |
0.76 – 0.85 |
Very good |
|
0.71 – 0.90 |
Very high |
0.66 – 0.75 |
Good |
|
0.51 – 0.70 |
High |
0.61 – 0.65 |
Satisfactory |
|
0.31 – 0.50 |
Moderate |
0.51 – 0.60 |
poor |
|
0.11 – 0.30 |
Low |
0.41 – 0.50 |
Bad |
|
< 0.10 |
Very low |
< 0.40 |
Very bad |
Specific points:
There are several typos which should be picked up at the copy-editing stage. However, there are a few which I was unable to decipher what was meant:
- P1 – line 37: unclear what is meant: “require horizontal positioning”. Does this mean that the instrument needs to be on a level plane in the horizontal?
P1 – line 38: the line “robustness from interferences” needs rephrasing. Something like free from interference sources such as shading by trees or buildings.
Replacement of “The measurement devices require horizontal positioning to the ground and robustness from interferences”
Addition of “The measurement devices require positioning in a plane level with the horizon and free of possible obstacles that act as sources of interference, providing shading on the device.”
- P1 - Line 41: not sure what is meant by the term “stroke”
P1 – line 42: “The Heliograph features a spherical lens” – I think it would be more correct to say: glass sphere.
Replace of “for direct measurement of SD stroke with greater acceptance and use in meteorological stations [2, 6, 7]. The Heliograph features a spherical lens that focuses the direct solar irradiance of the sun under”
Addition of “for direct measurement of SD with greater acceptance and use in meteorological stations [2, 6, 7]. The Heliograph features a glass sphere lens that focuses the direct solar irradiance of the sun under“
- P2 – line 50: “opening angle for capturing solar radiation below 30°” It would be more correct to say “an opening angle for capturing solar radiation <30° acceptance”
Replace of “has an opening angle for capturing solar radiation below 30°, causing the device to need adjustments”
Addition of “has a sensitivity angle to capture solar radiation with displacement less than 30°, making the device need adjustments”
- P2 – line 54: an unnecessary “-2”
Replace of “120 W.m-²-2”
Addition of “120 W.m-2”
- Line 216:
Replace of “Brazilian Agricultural Research Corporation (Embrapa)”
Addition of “Embrapa”.
- Line 276: This sentence reads as if it has been left in from the journal template!
Removing the expression “These metrics show the effectiveness and high performance of the proposed solution. This section may be divided by subheadings. It should provide a concise and precise description of the experimental results, their interpretation as well as the experimental conclusions that can be drawn.”
Reviewer 2 Report
In “Development and Validation of an Autonomous System for Measurement of Sunshine Duration” by da Rocha et al. a new electronic system for Sunshine Duration measuring is proposed. It is based on a shading pattern to detect the rapid transition of state of light. The proposed system is low-cost in comparison to other known Sunshine Duration measurements. Also authors pretend that it has no depency on the geographical parameters of the region. For the validation of the proposed system, experimental results were compared for the period October 2017 to July 2018 with the officials provided by INMET for Campina Grande in Brazil. Results from the comparison show an accuracy greater than 90%. The proposed device has been registered in the National Institute of Industrial Property. In my opinion, the paper is well written, the device is clearly described and the results are promising. This is why I think that it should be published in Sensors.
Author Response

(The authors gave the same response as above.)

Reviewer 3 Report
Review of Development and Validation of an Autonomous System for Measurement of Sunshine Duration
Authors: Rocha et al.
Overview
The authors present work on a low-cost ($150) instrument to measure Sunshine Duration (SD). There is a favourable intercomparison with other, more expensive ($2000) standard instruments, which is demonstrated statistically.
I have a few observations to make about the paper:
The method
- The authors present a contrast method in order to determine the sunshine duration, based on a quadrant of 4 photoresistor elements separated by orthogonal screens. I realise that this instrument has been deployed in Brazil, but it would be good if the authors could discuss the more global applicability of this instrument and method. As far as I can tell from the instrument setup and design, it might be problematic to deploy this instrument in high latitude regions, particularly in the winter months as the screens would likely completely shade the detectors when the solar angle is low. The authors could address this issue with data from early morning / late afternoon.
- There are other commercially available instruments which use a differential shading pattern, such as the SPN1 (https://www.delta-t.co.uk/product/spn1/) with associated publications which the authors may want to refer to (e.g. Wood et al., 2017 - http://plymsea.ac.uk/id/eprint/7456/6/amt-10-1723-2017.pdf). These would provide a good contrast with the authors instrument on grounds of cost, but that they provide a similar type of technique and robustness.
- Line 174: More details are required about the linguistic guidelines and how these were determined, rather than by saying that they were determined “experimentally”. This could mean anything.
- Line 25: it appears that Willmott’s confidence index is only mentioned here. It could be shortened to confidence index.
- I was entertained by the descriptions given in Table 1 as “sufferable” “bad” and “terrible”. However, I am not sure that such qualitative terms really belong in a scientific paper(!)
Specific points
There are several typos which should be picked up at the copy-editing stage. However, there are a few which I was unable to decipher what was meant:
- P1 – line 37: unclear what is meant: “require horizontal positioning”. Does this mean that the instrument needs to be on a level plane in the horizontal?
- P1 – line 38: the line “robustness from interferences” needs rephrasing. Something like free from interference sources such as shading by trees or buildings.
- P1 - Line 40: not sure what is meant by the term “stroke”
- P1 – line 41: “The Heliograph features a spherical lens” – I think it would be more correct to say: glass sphere.
- P2 – line 49: “opening angle for capturing solar radiation below 30°” It would be more correct to say “an opening angle for capturing solar radiation <30° acceptance”
- P2 – line 52: an unnecessary “-2”
- Line 261 – 263: This sentence reads as if it has been left in from the journal template!
Author Response

(The authors gave the same response as above.)
